# New Flexible Analogues of 8-Aza-7-deazapurine Nucleosides as Potential Antibacterial Agents

**DOI:** 10.3390/ijms242015421

**Published:** 2023-10-21

**Authors:** Anastasia Khandazhinskaya, Barbara Eletskaya, Anton Mironov, Irina Konstantinova, Olga Efremenkova, Sofya Andreevskaya, Tatiana Smirnova, Larisa Chernousova, Evgenia Kondrashova, Alexander Chizhov, Katherine Seley-Radtke, Sergey Kochetkov, Elena Matyugina

**Affiliations:** 1Engelhardt Institute of Molecular Biology, Russian Academy of Sciences, Vavilov St. 32, 119991 Moscow, Russia; khandazhinskaya@bk.ru (A.K.); evgenya.condrashova@gmail.com (E.K.); snk1952@gmail.com (S.K.); 2Shemyakin-Ovchinnikov Institute of Bioorganic Chemistry, Russian Academy of Sciences, Miklukho-Maklaya St. 16/10, 117997 Moscow, Russia; fraubarusya@gmail.com (B.E.); information@rudn.ru (A.M.); kid1968@yandex.ru (I.K.); 3Institute of Biochemical Technology and Nanotechnology, Peoples’ Friendship University of Russia Named after Patrice Lumumba, Miklukho-Maklaya St. 6, 117198 Moscow, Russia; 4Gause Institute of New Antibiotics, Bol’shaya Pirogovskaya St. 11, 119021 Moscow, Russia; ovefr@yandex.ru; 5Central Tuberculosis Research Institute, 2 Yauzskaya Alley, 107564 Moscow, Russia; andsofia@mail.ru (S.A.); s_tatka@mail.ru (T.S.); cniit@ctri.ru (L.C.); 6Zelinsky Institute of Organic Chemistry, Russian Academy of Sciences, Leninsky pr. 47, 119991 Moscow, Russia; chizhov@ioc.ac.ru; 7Department of Chemistry & Biochemistry, University of Maryland, Baltimore County, Baltimore, MD 21250, USA; kseley@umbc.edu

**Keywords:** fleximer, nucleoside analogues, antibacterial activity, inhibitor, antituberculosis activity

## Abstract

A variety of ribo-, 2′-deoxyribo-, and 5′-norcarbocyclic derivatives of the 8-aza-7-deazahypoxanthine fleximer scaffolds were designed, synthesized, and screened for antibacterial activity. Both chemical and chemoenzymatic methods of synthesis for the 8-aza-7-deazainosine fleximers were compared. In the case of the 8-aza-7-deazahypoxanthine fleximer, the transglycosylation reaction proceeded with the formation of side products. In the case of the protected fleximer base, 1-(4-benzyloxypyrimidin-5-yl)pyrazole, the reaction proceeded selectively with formation of only one product. However, both synthetic routes to realize the fleximer ribonucleoside (**3**) worked with equal efficiency. The new compounds, as well as some 8-aza-7-deazapurine nucleosides synthesized previously, were studied against Gram-positive and Gram-negative bacteria and *M. tuberculosis*. It was shown that 1-(β-D-ribofuranosyl)-4-(2-aminopyridin-3-yl)pyrazole (**19**) and 1-(2′,3′,4′-trihydroxycyclopent-1′-yl)-4-(pyrimidin-4(3H)-on-5-yl)pyrazole (**9**) were able to inhibit the growth of *M. smegmatis* mc2 155 by 99% at concentrations (MIC_99_) of 50 and 13 µg/mL, respectively. Antimycobacterial activities were revealed for 4-(4-aminopyridin-3-yl)-1H-pyrazol (**10**) and 1-(4′-hydroxy-2′-cyclopenten-1′-yl)-4-(4-benzyloxypyrimidin-5-yl)pyrazole (**6**). At concentrations (MIC_99_) of 40 and 20 µg/mL, respectively, the compounds resulted in 99% inhibition of *M. tuberculosis* growth.

## 1. Introduction

To date, great success has been achieved in the treatment of infectious diseases through antibiotic therapies [1]. Despite this, the rapid development of drug-resistant strains [2] of pathogenic microorganisms creates a need for the new and more effective antibiotics. The problem of drug resistance is more acute than ever in the case of tuberculosis, as it is one of the factors that reduces the effectiveness of many treatments [3,4].

In 2021, 10.6 million people were diagnosed with tuberculosis worldwide, and a total of 1.6 million deaths (including 187,000 people living with HIV) [4] have been attributed to it. Tuberculosis is the 13th leading cause of death globally and is the second leading cause of infectious death after COVID-19 (ahead of HIV/AIDS) [5]. The high incidence of tuberculosis in the global population is a serious threat to society [4]. In addition, over the past few years, the world has faced unprecedented challenges associated with the emergence of new pathogens such as SARS-CoV-2 [6], among other emerging and reemerging infectious diseases. This underscores the critical need for new and more effective therapeutic drugs. 

During decades of searching for molecules active against new and emerging bacterial infections, drugs belonging to various chemical classes of compounds have been discovered. Unfortunately, again, the development of resistance has limited their utility [7]. One potential solution is to explore new viral and bacterial targets, as well as classes of drugs not previously considered [8]. Another is the development of compounds that exhibit broad-spectrum activity, or the use of combination therapies, thereby reducing susceptibility to the development of resistance [9].

Nucleoside analogues, which are traditionally considered as antiviral [10,11,12] or antitumor agents [13], have also recently become attractive as potential antibacterial compounds, including as a part of a repurposing strategy [14]. For example, the well-known anti-HIV drug zidovudine (AZT) [15] has been demonstrated to have potent activity against many pathogenic Gram-negative bacteria, including *Escherichia coli*, *Salmonella typhimurium*, *Klebsiella pneumoniae*, *Shigella flexneri*, and *Haemophilus influenzae* as well as isolates that are resistant to conventional antibiotics [14]. Pyrimidine nucleosides, bearing long substituents in the 5-position of the heterocyclic base, were able to inhibit *Mycobacterium tuberculosis*, *Mycobacterium bovis*, and *Mycobacterium avium* [16,17,18,19,20,21,22,23,24]. A group of 5-substituted uridine analogues inhibited the mycobacteria growth including MDR (multidrug-resistant) strains in the low micromolar range [20,21]. 

Among the purine nucleosides, there are also compounds which have exhibited antibacterial properties. For example, there are the compounds developed by the Aldrich group, who synthesized a number of adenosine derivatives which proved to be inhibitors of *M. tuberculosis* [25,26]. Moreover, analogues of 8-aza-, 7-deaza-, 9-deaza-, and 8-aza-7-deazapurines showed pronounced inhibitory properties against *M. tuberculosis* [27,28,29]. 

In terms of the search for new compounds with broad spectrum activity, a very promising group of purine nucleoside analogues are the fleximers [30,31]. It has been shown that these unique compounds, where the purine base is split into its two separate components, have exhibited a wide spectrum of biological activity [30]. The additional flexibility of such nucleoside analogues is a significant advantage and in the case of resistant pathogens makes it possible for a potential drug to “adjust” to the binding site containing point mutations and retain inhibitory activity [30]. 

In this work, we present the synthesis of several new fleximer ribo-, 2′-deoxyribo-, and 5′-norcarbocyclic derivatives of the 8-aza-7-deazahypoxanthine scaffold and the testing of 8-aza-7-deazapurine nucleoside analogues against Gram-positive and Gram-negative bacteria and mycobacteria.

## 2. Results

### 2.1. Synthesis of the Target Compounds

We have previously compared the chemical and chemoenzymatic methods for the synthesis of fleximer adenosine analogues [32]. It was shown that fleximer analogues of 8-aza-7-deazaadenine are useful substrates for purine nucleoside phosphorylase *E. coli* (PNP *E. coli*) in the synthesis of corresponding modified nucleosides. Fleximer 8-aza-7-deazaadenine ribonucleosides can be obtained through both methods with comparable effectiveness. In contrast, the chemical synthesis of 2′-deoxynucleoside analogues resulted in mixtures of α- and β-anomers, so an additional purification step is needed. Thus, enzymatic transglycosylation proved to be the preferred route for the 2′-deoxyribonucleosides [32].

In an effort to explore these routes for other fleximer aza/deaza purines, we have compared the chemical and chemo-enzymatic methods of 8-aza-7-deazainosine fleximer synthesis. In that regard, a series of experiments was carried out to determine the substrate specificity of *E. coli* purine nucleoside phosphorylase to heterocyclic bases **1a** and **1b**. In the case of **1a**, the transglycosylation reaction (Figure 1) proceeded with the formation of side products for both the riboside and 2′-deoxyriboside (see Appendix A). This occurs due to the availability of several glycosylation sites. Base **1a** contains four nitrogen atoms and lacks functional group protection. In the case of the fleximer base **1b**, the reaction proceeds selectively with the formation of just one product, as observed by chromatography (see Appendix A). The presence of benzyl protecting groups creates steric hindrance for the enzyme for glycosylation of the pyrimidine, thereby ensuring the regioselectivity of the reaction. Thus, we chose to use the chemoenzymatic glycosylation method for the synthesis of the nucleoside analogues of the 8-aza-7-deazahypoxanthine fleximer using the protected flex-base **1b**.

Protype reactions were carried out to determine the best donor of ribose and 2′-deoxyribose for base **1b**. In the first series of experiments, uridine, guanine, adenine, and inosine were chosen as the potential carbohydrate donors. In the second series, 2′-deoxyuridine, 2′-deoxyguanosine, 2′-deoxyadenosine, and 2′-deoxyinosine were chosen. The reactions were carried out in a phosphate buffer pH 7.0 at 50 °C. Interestingly, there was no difference in glycosylation efficiency when the natural purine nucleosides inosine, guanosine, and adenosine were used as donors. However, for uridine the formation of the target nucleoside proceeded faster (conversion 97% after 1 h). As a result, uridine and 2′-deoxyuridine, respectively, were chosen as the ribose/deoxyribose donors. 

Nucleoside analogues **2** and **3** were then synthesized chemoenzymatically using uridine phosphorylase (UP *E. coli*), PNP *E. coli*, and 1-(4-benzyloxypyrimidin-5-yl)pyrazole **1b** as the base. Removal of the benzyl protective group by a hydrogenation reaction on a palladium catalyst (Figure 1) afforded fleximer nucleosides **4** and **5** in 25–26% yield.

For chemical synthesis of fleximer ribonucleoside **5**, protected flex-base **1b** was also chosen (Figure 1). Using classical Vorbruggen coupling methodology [33], compound **1b** was refluxed in hexamethyldisilazane, followed by glycosylation with β-D-ribofuranose 1,2,3,5-O-tetraacetate in the presence of trimethylsilyl trifluoromethanesulfonate. Protected β-D-ribonucleoside **3** was obtained as the main product with 75% yield. Removal of the acetyl and benzyl protecting groups led to the target fleximer analogue **5** (18% yield).

So, in the case of the 8-aza-7-deazainosine fleximer **3**, synthesis started from 1-(4-benzyloxypyrimidin-5-yl)pyrazole **1b**, and both methods worked with equal efficiency. Chemoenzymatic synthesis gave compound **3** in 77% yield and chemical synthesis in 71% yield.

5′-Norcarbocyclic derivatives of fleximer 8-aza-7-deazahypoxantine were also synthesized from 1-(4-benzyloxypyrimidin-5-yl)pyrazole **1b** and the known 5′-norcarbocyclic precursor 6-oxobicyclo[3.1.0.]hex-2-ene [34] using the Trost condensation protocol [35].

The 5′-norcarbocyclic analogue of 8-aza-7-deazainosine **9** was synthesized by oxidation of the double bond of compound **6** with osmium tetraoxide in dioxane–water (10:1), and subsequent removal of the benzyl protecting group of **8** with palladium on carbon under a hydrogen atmosphere (Figure 2). Product **9** was obtained in a 38% yield in two steps. An attempt to directly remove the benzyl group from compound **6** through hydrogenation produced compound **7** in a 63% yield. 

### 2.2. Antimicrobial Studies

In order to evaluate the antimicrobial effect of the fleximers, heterocyclic base analogues **1a** and **1b**, the new derivatives of 8-aza-7-deazahypoxanthine fleximers **2**–**9**, and the previously reported 8-aza-7-deazapurine fleximer nucleoside analogues **10**–**22** [32,36,37] (Figure 1), were tested against a number of microorganisms.

The antimicrobial activity of the compounds was studied as previously described [38] by measuring their ability to inhibit the growth in vitro of Gram-positive and Gram-negative bacteria as well as fungi. Among the strains tested were *Bacillus subtilis* ATCC 6633; methicillin-resistant *Staphylococcus aureus* strain INA 00761 (MRSA); methicillin-sensitive *Staphylococcus aureus* FDA 209P (MSSA); *Micrococcus luteus* NCTC 8340; vancomycin-resistant *Leuconostoc mesenteroides* VKPM B-4177 (VKPM); *M. smegmatis* mc2 155; *Escherichia coli* ATCC 25922; *Pseudomonas aeruginosa* ATCC 27853 (causes nosocomial infections often refractory to antibiotic therapy due to multidrug resistance); baking yeast *Saccharomyces cerevisiae* INA 01129; fungal test culture *Aspergillus niger* INA 00760. The results revealed that only 1-(β-D-ribofuranosyl)-4-(2-aminopyridin-3-yl)pyrazole (**19**) and 1-(2′,3′,4′-trihydroxycyclopent-1′-yl)-4-(pyrimidin-4(3H)-on-5-yl)pyrazole (**9**) inhibited the growth of *M. smegmatis* mc2 155 by 99% at concentrations (MIC_99_) of 50 and 13 µg/mL, respectively.

The ability of the fleximer analogues to inhibit the growth of *M. tuberculosis* H37Rv was also tested. The growth of *M. tuberculosis* laboratory strain H37Rv was exposed to the compounds at concentrations of 10–40 µg/mL; these were identical to those observed in the control group (culture, growing on the medium without the tested compound). Only 4-(4-aminopyridin-3-yl)-1H-pyrazol (**10**) at concentrations (MIC_99_) of 40 µg/mL and 1-(4′-hydroxy-2′-cyclopenten-1′-yl)-4-(4-benzyloxypyrimidin-5-yl)pyrazole (**6**) at concentrations of 20 µg/mL caused 99% inhibition of bacterial growth. Compound (**6**) completely suppressed the growth of the culture at a concentration of 40 μg/mL.

## 3. Discussion

In this work, we continue to study the importance and biological scope of these flexible nucleoside analogues known as fleximers. Earlier [32], we have shown that fleximer analogues of 8-aza-7-deazaadenine are useful substrates for purine nucleoside phosphorylase *E. coli* (PNP *E. coli*). It also provided the opportunity to further investigate and compare the two routes of synthesis using chemical methods and enzymatic transglycosylation, in this case for the 8-aza-7-deazainosine fleximer scaffold. It was found during the experiments, through the determination of the substrate specificity of *E. coli* PNP to heterocyclic bases **1a** and **1b**, that the benzyl protective group is required for ensuring the regioselectivity of the reaction. As a result, protected flex-base **1b** was chosen for enzymatic glycosylation as well as for the chemical synthesis of ribonucleoside **3** using the classical Vorbruggen procedure. In the case of the synthesis of the 8-aza-7-deazainosine fleximer **3** starting from 1-(4-benzyloxypyrimidin-5-yl)pyrazole **1b**, both methods worked with equal efficiency. 

In addition to fleximers with traditional ribose or 2′-deoxyribose, we synthesized derivatives of 8-aza-7-deazahypoxanthine bearing a 5′-norcarbocyclic fragment as the sugar moiety. 5′-Norcarbocyclic nucleoside analogues [39] have a number of advantages. One of them is the absence of 5′-CH_2_ group, which prevents phosphorylation thereby resulting in decreasing cytotoxicity. At the same time, biological properties not associated with phosphorylation are retained. This feature also helps to exclude classical nucleoside polymerases inhibition [40], but at the same time several 5′-norcarbocyclic nucleoside analogues have been shown to act as nonnucleoside inhibitors of viral RNA polymerases [41] or reverse transcriptase [42,43].

Fleximer heterocyclic base analogues, derivatives of 8-aza-7-deazahypoxanthine fleximers as well as the 8-aza-7-deazapurine fleximer analogues were also studied against a number of microorganisms.

Antibacterial screening has shown that flexible analogues of 8-aza-7-deazapurine nucleosides, compounds **9** and **19**, inhibited the growth of *M. smegmatis* mc2 155 in 13 and 50 µg/mL, respectively. Analogues **6** and **10** caused 99% inhibition of *M. tuberculosis* H37Rv, at concentrations of 20 and 40 µg/mL, respectively. It was not surprising that different compounds were active against *M. smegmatis* and *M. tuberculosis*, since they are not closely related groups of mycobacteria. *M. tuberculosis* is slow-growing mycobacteria, and *M. smegmatis* is fast-growing one, so these two types of microorganisms may have differential sensitivities to the various fleximer analogues. Nevertheless, the data obtained will help guide for further optimization of new fleximer scaffolds in the search for new and more effective antibacterial agents.

## 4. Materials and Methods

### 4.1. Chemistry

Commercial reagents for reactions (Acros, Aldrich, Thermo Fisher Scientific, Tokyo Chemical Industry, and Fluka) were used without purification; anhydrous solvents were purified according to standard procedures. Column chromatography was performed on Silica Gel 60 0.040–0.063 mm (Merck, Darmstadt, Germany) columns, Dowex-50 (H+). Preparative liquid chromatography (PLC) was performed on Silica Gel 60 F254 with concentrating zone glass plates (Merck, Germany). Thin layer chromatography (TLC) was performed on Silica Gel 60 F254 aluminum-backed plates (Merck, Germany).

NMR spectra were recorded on Bruker Avance III 300 spectrometer (Bruker BioSpin, Rheinstetten, Germany) or Bruker Avance II 700 spectrometer (Bruker BioSpin, Rheinstetten, Germany) in CDCl_3_, CD_3_OD, or DMSO-d6 at 30 °C.

Liquid chromatography mass spectrometry was performed using an Agilent 6210 TOF LC–MS system (Agilent Technologies, Santa Clara, CA, USA). 

The UV spectra were recorded using a Beckman DU-530 spectrophotometer (Beckman Coulter Inc., Brea, CA, USA)

Analytical HPLC was performed using the Waters system (Waters 1525, Waters 2489, Breeze 2, (Waters Inc., Milford, MA, USA); column Supelco Ascentis^®^ Express C18, 2.7 μm 7.5 × 3.0 mm, eluent A—0.1% TFA/H_2_O, eluent B—70% acetonitrile in 0.1% TFA/H_2_O, flow rate 0.5 mL/min, detection at 280 nm. Gradient 0–50% B, 20 min.

High-resolution mass spectra (HRMS) were obtained on a Bruker Daltonics micrOTOF II instrument using electrospray ionization (ESI). The measurements were acquired in a positive ion mode with the following parameters: interface capillary voltage–4500 V; mass range from *m*/*z* 50 to 3000; internal calibration (ESI Tuning Mix, Agilent); nebulizer pressure—0.3 Bar; flow rate—3 µL/min; dry gas nitrogen (4.0 L/min); interface temperature was set at 180 °C. Syringe injection was used. 

**1-(4-benzyloxypyrimidin-5-yl)pyrazole** (**1b**). To 5-Bromo-4-benzyloxypyrimidine (1.3 g, 5 mmol) in 1,2-dimethoxyethane (100 mL), tetrakis(triphenylphosphine)palladium (5 mol %) was added under argon atmosphere and stirred for 15 min. Then, 4-(4,4,5,5-tetramethyl-1,3,2-dioxoboran-2-yl)-1H-pyrazole (1.1 g, 5.5 mmol) in 1,2-dimethoxyethane (15 mL) and a saturated aq. solution of sodium bicarbonate (10 mL) were added. The reaction mixture was refluxed at 90 °C for 4 h, then concentrated in vacuo and partitioned between water (30 mL) and chloroform (100 mL). The organic layer was washed with brine, dried (Na_2_SO_4_), concentrated in vacuo, and the residue was purified by column chromatography on silica gel in CH_3_Cl: MeOH (95:5) system to give flexible base **1b** (795 mg) in 63% yield. ^1^H NMR (300 MHz, CD_3_OD) δ: 8.41 (s, 1H, H-2B), 8.27 (s, 1H, H-5A), 8.24 (s, 2H, H-6B, H-3A), 7.30–7.40 (m, 5H, benzyl group), 5.24 (s, 2H, CH_2_) ppm. ^13^C NMR (75.5 MHz, CD_3_OD) δ: 159.3, 148.9, 147.95, 145.6, 142.1, 134.9, 132.6, 128.9, 128.4, 127.9, 120.5, 113.1, 50.4, 24.1 ppm. HRMS, *m/z*: calculated for C_12_H_14_N_4_O [M + H]^+^ 253.1084, found [M + H]^+^ 253.1088. UV λmax 310 nm.

**1-(4-pyrimidin-4(3H)-on-5-yl)pyrazole** (**1a**). Deprotection of compound **1b** (200 mg, 0.8 mmol) was carried out in methanol (10 mL) with addition of 10% Pd/C (30 mg) and TFA (2 mL) under an H_2_ atmosphere (1 bar). The reaction mixture was stirred for 12 h. The solvent was removed in vacuo and the residue was purified on a silica gel column to give compound **1a** (50 mg) in 40% yield. ^1^H NMR (300 MHz, DMSO-d6) δ: 8.38 (s, 2H, H-2B, H-5A), 8.06 (s, 2H, H-6B, H-3A) ppm. ^13^C NMR (75.5 MHz, DMSO-d6) δ: 148.1, 147.4, 137.2, 127.7, 126.3, 120.8, 113.5 ppm. HRMS, *m*/*z*: calculated for C_7_H_6_N_4_O [M + H]^+^ 163.0614, found [M + H]^+^ 163.0593. UV λmax 310 nm.

### 4.2. General Procedure for the Enzymatic Synthesis of Fleximer Nucleosides

The flex-base **1b** and 2′-deoxyuridine/uridine at ratios of 1:2 were dissolved in 70 mL 10 mM potassium phosphate buffer (pH 7.0) at 40–50 °C. The enzymes 3.2 e.u./mL PNP and 4.0 e.u./mL UP *E. coli* were added. The reaction mixtures were incubated at 50 °C until the conversion reached 98–100%, according to the RP-HPLC data. When the conversion reached the maximal value, the reaction was terminated by ultrafiltration using Amicon^®^ Ultra-4 Centrifugal Filter Unit (10 kDa, Merck Millipore, Darmstadt, Germany) and concentrated in vacuo to a minimum volume (5 mL). The isolation of the fleximer nucleosides **2**–**3** was performed using reverse-phase column chromatography (silica gel C18, Merck), using a column of 20 mm × 550 mm. Nucleosides were eluted from the column with gradient eluent A 100% water–eluent B 50:50% ethanol/water.

**1-(β-D-2′-Deoxyribofuranosyl)-4-(4-benzyloxypyrimidin-5-yl)pyrazole** (**2**). ^1^H NMR (700 MHz, DMSO-d6) δ: 8.61 (s, 1H, H-2B), 8.52 (s, 1H, H-5A), 8.42 (s, 1H, H-6B), 8.12 (s, 1H, H-3A), 7.36 and 7.30 (m, 4H and m, 1H, Ph), 6.13 (t, J = 6.32 Hz, 1H, H-1′), 5.22 (br.s, 0.78 H, 5′-OH), 5.20 (s, 2H, -CH_2_-), 4.80 (br.t, 0.78 H, 3′-OH), 4.36 (br.s, 0.78 H, H-3′), 3.82 (m, 1 H, H-4′), 3.51 (m, 1H, Ha-5′), 3.40 (m, 1H, Hb-5′), 2.59 (m, 1H, Ha-2′), 2.29 (m, 1H, Ha-2′) ppm. ^13^C NMR (176 MHz, DMSO-d6) δ: 158.57, 149.66, 147.20, тa137.54, 136.81, 128.51, 127.62, 128.57, 119.04, 114.54, 88.95, 87.67, 70.79, 62.10, 49.82, 39.94 ppm. ^15^N NMR (71 MHz, DMSO-d6) δ: 301.30 (N2A), 244.30 (N1B), 226.43 (N1A), 186.32 (N3B) ppm. HRMS, *m*/*z*: calculated for C_19_H_20_N_4_O_4_, [M + H]^+^ 369.1557, found [M + H]^+^ 369.1564. Yield 71.6%, 33.4 mg. Purity 99.54%. Rt 8.33 min. UV λmax 305, 299, 253 nm, ε 10550 (299 nm). 

**1-(β-D-Ribofuranosyl)-4-(4-benzyloxypyrimidin-5-yl)pyrazole** (**3**). ^1^H NMR (700 MHz, DMSO-d6) δ: 8.62 (s, 1H, H-2B), 8.56 (s, 1H, H-5A), 8.42 (s, 1H, H-6B), 8.14 (s, 1H, H-3A), 7.36 and 7.31 (m, 4H and m, 1H, Ph), 5.69 (d, J = 4.43 Hz, 1H, H-1′), 5.34 (d, J = 5.92 Hz, 1H, 2′-OH), 5.20 (s, 2H, -CH_2_-), 5.08 (d, J = 5.48 Hz, 1H, 3′-OH), 4.84 (t, J = 5.66 Hz, 1H, 5′-OH), 4.34(q, J = 4.98; 5.66 Hz, 1H, H-2′), 4.12 (q, J = 4.97; 5.16 Hz, 1H, H-3′), 3.91 (q, J = 4.85; 4.58 Hz, 1H, H-4′), 3.59 (m, 1H, Ha-5′), 3.48 (m, 1H, Hb-5′), ppm. ^13^C NMR (176 MHz, DMSO-d6) δ: 158.92, 150.30, 147.83, 138.35, 136.94, 129.26, 128.20, 129.12, 119.63, 114.54, 93.80, 85.58, 74.98, 71.05, 62.40. 49.82 ppm. ^15^N NMR (71 MHz, DMSO-d6) δ: 302.61 (N2A), 244.33 (N1B), 222.35 (N1A) 186.45(N3B) ppm. HRMS, *m*/*z*: calculated for C_19_H_20_N_4_O_5_, [M + H]^+^ 385.1506, found [M + H]^+^ 385.1539. Yield 76.6%, 40.9 mg. Purity 97.03%. Rt 7.78 min. UV λmax 305, 299, 253 nm, ε 9890 (299 nm).

### 4.3. The Palladium-on-Carbon (Pd/C)-Catalyzed Hydrogenative Deprotection of the N-Benzyl-Protecting Group

Protected fleximer nucleosides **2**–**3** (10 mg, 0.027 mmol) were dissolved in ethanol (2 mL). The solution diluted with ethanol (10 mL), and 10% Pd/C (10 mg) was added. The reaction mixture was stirred under an H_2_ atmosphere (1 bar) for 12 h. Aliquots (60 μL) were taken from the reaction mixture, and the progress of the reactions was monitored using HPLC. When the conversion reached the maximal value, the reaction mixture was passed through a filter, and the filtrate was concentrated in vacuo. Isolation of the fleximers nucleosides **4**–**5** was performed using Phenomenex Strata C18-E cartridge (200 mg/3 mL). Nucleosides were eluted from the column with gradient eluent A 100% water–eluent B 50:50% ethanol/water.

**1-(β-D-2′-Deoxyribofuranosyl)-4-(pyrimidin-4(3H)-on-5-yl)pyrazole** (**4**). ^1^H NMR (700 MHz, DMSO-d6) δ: 8.51 (s, 1H, H-5A), 8.38 (s, 1H, H-6B), 8.11 (s, 1H, H-3A), 8.09 (s, 1H, H-2B), 6.14 (t, J = 6.34 Hz, 1H, H-1′), 5.22 (d, J = 4.36 Hz), 4.80 (t, J = 5.59 Hz, 1H, 5′-OH), 4.36 (m, 1H, H-2′), 4.34 (m, 1H, H-3′), 3.83 (m, 1H, H-4′), 3.52 (m, 1H, Ha-5′), 3.42 (m,1H, Hb-5′), 2.61 (m, 1H, Ha-2′), 2.24 (m, 1H, Ha-2′) ppm. Yield 26%, 2 mg. Purity 96.84%. Rt 4.29 min. HRMS, *m*/*z*: calculated for C_12_H_14_N_4_O_4_ [M + H]^+^ 279.1088, found [M + H]^+^ 279.1103. UV λmax 305, 298, 251 nm.

**1-(β-D-Ribofuranosyl)-4-(pyrimidin-4(3H)-on-5-yl)pyrazole** (**5**). ^1^H NMR (700 MHz, DMSO-d6) δ: 8.53 (s, 1H, H-5A), 8.32 (s, 1H, H-6B), 8.10 (s, 1H, H-3A), 8.07 (s, 1H, H-2B), 5.68 (t, J = 5.68 Hz, 1H, H-1′), 4.86 (br.s, 0.55 H, 5′-OH), 4.35 (t, J = 4.41 Hz, 1H, H-2′), 4.13 (t, J = 4.89 Hz, 1H, H-3′), 3.90 (q, J = 4.72; 4.79 Hz, 1H, H-4′), 3.59 (m, 1H, Ha-5′), 3.49 (m,1H, Hb-5′). ^13^C NMR (176 MHz, DMSO-d6) δ: 161.70 (C4B), 148.92 (C2B), 147.75 (C6B), 137.52 (C3A), 128.21 (C5A), 118.59 (C4A), 114.91 (C5B), 93.78 (C1′), 84.95 (C4′), 74.35 (C2′), 70.48 (C3′), 61.89 (C5′). ^15^N NMR (71 MHz, DMSO-d6) δ: 300.79 (N2A), 240.54 (N1B), 221.45 (N1A), 194.27 (N3B) ppm. HRMS, *m*/*z*: calculated for C_12_H_14_N_4_O_5_ [M + H]^+^ 295.1037, found [M + H]^+^ 295.1028. Yield 25%, 2 mg. Purity 97.16%, Rt 4.38 min. UV λmax 305, 298, 251 nm.

### 4.4. Chemical Synthesis of Fleximer Nucleosides

**1-(β-D-(2′,3′,5′-Triacetylribofuranosyl))-4-(4-benzyloxypyrimidin-5-yl)pyrazole.** To 1-(4-benzyloxypyrimidin-5-yl)pyrazole **1b** (100 mg, 0.6 mmol) (NH_4_)_2_SO_4_ (10 mg), HMDS (20 mL) and Py (2 mL) were added. The reaction mixture was refluxed for 3 h at 150 °C. The solvent then was concentrated and the residue dissolved in acetonitrile. β-D-Ribofuranose-1,2,3,5-tetraacetate (150 mg, 0.6 mmol) and Trf were added and left overnight. The solvent was evaporated and purification by preparative chromatography in Hex:EtOAc + MeOH (1:3 + 1%) gave the product as a white powder (170 mg, yield 75%). ^1^H NMR (300 MHz, CDCl_3_) δ: 9.03 (s, 1H, H-6B), 8.52 (s, 1H, H-2B), 8.21 (d, J = 15.9 Hz, 1H, H-3A), 8.01 (s, 1H, H-5A), 7.43 (m, 5H, Ph), 5.94 (d, J = 3.3 Hz, 1H, H-1′), 5.79 (dd, J = 5.2, 3.4 Hz, 1H, H-2′), 5.68 (d, J = 5.3 Hz, 1H, H-3′), 5.33 (s, 2H -CH_2_-), 4.40–4.43 (m, 3H, Ac-5′), 4.17–4.25 (m, 3H, H-5′), 2.11 (s, 6H, 2xAc), 1.37 (m, 4H, H-4′) ppm. ^13^C NMR (75 MHz, CDCl_3_) δ: 233.8, 225.1, 222.3, 199.6, 182.0, 170.7, 169.4, 158.0, 148.6, 140.6, 138.6, 134.1, 130.3, 129.9, 129.2, 128.5, 113.8, 91.6, 80.2, 77.0, 76.6, 74.4, 71.0, 63.4, 51.2, 21.1 ppm.

**1-(β-D-Ribofuranosyl)-4-(4-benzyloxypyrimidin-5-yl)pyrazole (3).** 1-(β-D-(2′,3′,5′-Triacetylribofuranosyl))-4-(4-benzyloxypyrimidin-5-yl)pyrazole (100 mg, 0.2 mmol) was dissolved in 7N ammonia in methanol (20 mL). The reaction mixture was kept at 36 °C for 3 h. Purification with preparative chromatography on silica gel glass plate in chloroform/methanol (95:5) system, gave the riboside (**3**) as a white powder (71 mg) with 95% yield. 

**1-(β-D-Ribofuranosyl)-4-(pyrimidin-4(3H)-on-5-yl)pyrazole (5).** Riboside (**3**) (50 mg, 0.13 mmol) was dissolved in methanol (10 mL), and 10% Pd/C (30 mg) and TFA (2 mL) were added. The reaction mixture was stirred under an H_2_ atmosphere (1 bar) for 12 h. After completion of the reaction, the mixture was filtered through a pad of Celite and the filtrate evaporated to dryness. The residue was purified via a silica gel column eluting with chloroform/methanol (8:2) to give compound **5** as a white powder (32 mg, 18% on 3 steps).

**1-(4′-Hydroxy-2′-cyclopenten-1′-yl)-4-(4-benzyloxypyrimidin-5-yl)pyrazole** (**6**). 1-(4-Benzyloxypyrimidin-5-yl)pyrazole **1** (250 mg, 1 mmol) were dissolved in DMF and re-evaporated 2 times. Then, 6-oxybicyclo [3.1.0.]hex-2-ene (1.3 eq) in 2–3 mL THF and Pd(PPh_3_)_4_ 5 mol% were added. The reaction mixture was stirred for 18h and the solvents evaporated. The product was purified by column chromatography on silica gel eluting with chloroform/methanol (98:2) to give 254 mg of compound **6** in 75% yield. ^1^H NMR (300 MHz, CD_3_OD) δ: 8.39 (s, 1H, H-2B), 8.22 (s, 1H, H-5A), 8.12 (s, 1H, H-3A), 7.92 (s, 1H, H-6B), 7.30–7.44 (m, 5H, Ph), 6.31 dt, J = 5.6, 1.8 Hz, (1H, H-2′), 5.97 (dd, J = 5.5, 2.5 Hz, 1H, H-3′), 5.21–5.18 (m, 3H, CH_2_, H-1′), 4.75–4.77 (m, 1H, H-4′), 2.62–2.72 (m, 1H, H-5′a), 2.1–2.04 (m, 1H, H-5′b) ppm. ^13^C NMR (75.5 MHz, CDCl_3_) δ: 159.4, 148.0, 146.6, 139.2, 137.1, 129.3, 129.1(*2), 128.5(*2), 128.1(*2), 113.6, 74.7, 65.6, 50.3, 40.5 ppm. HRMS, *m*/*z*: calculated for C_19_H_18_N_4_O_2_ [M + H]^+^ 335.1503, found [M + H]^+^ 335.1500; calculated for C_19_H_18_N_4_O_2_ [M + Na]^+^ 357.1322, found [M + Na]^+^ 357.1316; calculated for C_19_H_18_N_4_O_2_ [M + K]^+^ 373.1061, found [M + K]^+^ 373.1053; UV λmax 315 nm.

**1-(4′-Hydroxycyclopent-1′-yl)-4-(pyrimidin-4(3H)-on-5-yl)pyrazole** (**7**). To a solution of 6 (50 mg, 0.15 mmol) in anhydrous MeOH (10 mL), 10% Pd/C (40 mg) was added under a H_2_ atmosphere, and the reaction mixture was stirred at room temperature under a H_2_ atmosphere for 18 h. The mixture was filtered through a pad of Celite and the filtrate evaporated to dryness. The residue was purified via a silica gel column eluting with chloroform/methanol (9:1) to give compound **7** as an off-white powder (23 mg, 63%). ^1^H NMR (300 MHz, DMSO-d6) δ: 12.62 (s, 1H, NH), 8.41 (s, 1H, H-2B), 8.35 (s, 1H, H-5A), 8.06 (s, 1H, H-3A), 8.03 (s, 1H, H-6B), 4.87 (d, J = 4.8 Hz, 1H, H-1′), 4.63–4.82 (m, 1H, OH), 4.16–4.21 (m, 1H, H-4′), 2.37–2.39 (m, 1H, H-5′a), 1.91–2.17 (m, 2H, CH_2_), 1.91–1.59 (m, 3H, CH_2_, H-5′b) ppm. ^13^C NMR (75.5 MHz, DMSO-d6) δ: 159.8, 147.8, 147.4, 136.8, 128.0, 120.6, 114.0, 70.8, 61.0, 42.2, 34.5, 31.3 ppm. HRMS, *m*/*z*: calculated for C_12_H_14_N_4_O_2_ [M + Na]^+^ 269.1009, found [M + Na]^+^ 269.1018. UV λmax 310 nm.

**1-(2′,3′,4′-Trihydroxycyclopent-1′-yl)-4-(4-benzyloxypyrimidine-5-yl)pyrazole** (**8**). Compound **6** (200 mg, 0.6 mmol) was dissolved in dioxane/water (10:1) system. NMMO (10 equiv.) and osmium tetroxide (0.25 equiv.) were added to the solution. The reaction mixture was stirred for 6 h and the solvents were evaporated. The residue was purified on a silica gel column eluting with chloroform/methanol (9:1) to give the product as a white powder in 89% yield. ^1^H NMR (300 MHz, CD_3_OD) δ: 8.39–8.45 (m, 2H, H-2B, H-5A), 8.32 (s, 1H, H-3A), 8.07 (s, 1H, H-6B), 7.28–7.41 (m, 5H, Ph), 5.25 (s, 2H, CH_2_), 4.63 (dt, J = 9.3, 7.7 Hz, 1H, H-2′), 4.43 (dd, J = 7.4, 5.1 Hz, 1H, H-3′), 4.06–4.10 (m, 1H, H-1′), 3.91–3.97 (m, 1H, H-4′), 2.67–2.78 (m, 1H, H-5′a), 1.93–2.02 (m, 1H, H-5′b) ppm. ^13^C NMR (75.5 MHz, CD_3_OD) δ: 159.4, 149.1, 146.5, 137.2, 135.9, 129.1, 128.5, 127.9, 127.7, 120.3, 113.7, 77.2, 76.2, 74.1, 65.9, 49.8, 36.1 ppm. HRMS, *m*/*z*: calculated for C_19_H_20_N_4_O_4_ [M + H]^+^ 369.1557, found [M + H]^+^ 369.1553; calculated for C_19_H_20_N_4_O_4_ [M + Na]^+^ 391.1377, found [M + Na]^+^ 391.1370; calculated for C_19_H_20_N_4_O_4_ [M + K]^+^ 407.1116, found [M + K]^+^ 407.1110. UV λmax 315 nm.

**1-(2′,3′,4′-Trihydroxycyclopent-1′-yl)-4-(pyrimidin-4(3H)-on-5-yl)pyrazole** (**9**). To a solution of **8** (100 mg, 0.27 mmol) in anhydrous MeOH (10 mL), 10% Pd/C (60 mg) was added and the reaction mixture was stirred at room temperature under a H_2_ atmosphere for 48 h. The mixture was filtered through a pad of Celite and the filtrate evaporated to dryness. The residue was purified via a silica gel column eluting with chloroform/methanol (8:2) to give compound **9** as a white powder (54 mg, 72%). ^1^H NMR (300 MHz, DMSO-d6) δ: 12.71 (s, 1H, NH), 8.32–8.44 (m, 2H, H-2B, H-5A), 7.86–8.14 (m, 2H, H-3A, H-6B), 5.02 (d, J = 4.2 Hz, 1H, OH), 4.91 (d, J = 6.8 Hz, 1H, OH), 4.82 (d, J = 3.8 Hz, 1H, OH), 4.44–4.56 (m, 1H, H-2′), 4.23–4.25 (m, 1H, H-3′), 3.83–3.94 (m, 1H, H-1′), 3.70–3.71 (m, 1H, H-4′), 1.75–1.84 (m, 1H, H-5′a), 0.92–0.94 (m, 1H, H-5′b) ppm. ^13^C NMR (75.5 MHz, DMSO-d6) δ: 158.6, 148.0, 137.0, 129.0, 119.7, 115.7, 77.4, 76.4, 73.8, 65.7, 37.0, 29.4 ppm. HRMS, *m*/*z*: calculated for C_12_H_14_N_4_O_4_ [M + H]^+^ 279.1088, found [M + H]^+^ 279.1081. UV λmax 310 nm.

### 4.5. Antimicrobial Activity

The antimicrobial activity of the compounds was determined by the method of two-fold serial dilutions in the #2 Gause medium of the following composition (g/L): glucose-10, peptone-5, tryptone-3, sodium chloride-5; tap water. Eight strains of bacteria and two strains of fungi were used as test cultures: *Bacillus subtilis* ATCC 6633, *Leuconostoc mesenteroides* VKPM B-4177, *Micrococcus luteus* NCTC 8340, *Staphylococcus aureus* FDA 209P (MSSA), *S. aureus* INA 00761 (MRSA), *Mycobacterium smegmatis* mc2 155, *Escherichia coli* ATCC 25922, *Pseudomonas aeruginosa* ATCC 27853, and fungi *Saccharomyces cerevisiae* RIA 259 and *Aspergillus niger* INA 00760. The inoculation of the medium was 106 CFU/mL. Fungi and *L. mesenteroides* VKPM B-4177 were incubated at 28 °C and the remaining bacterial strains were incubated at 37 °C. The duration of the incubation was 24 h except for the *A. niger* INA 00760 and *M. smegmatis* mc2 155, which were cultivated for two days. MIC corresponded to the complete absence of growth. As a control, test strains were incubated in a medium without test substances, as well as in a medium with the addition of a solvent (methanol at a maximum concentration of 3 vol.%).

### 4.6. Antituberculosis Tests

The virulent laboratory strain *M. tuberculosis* H37Rv was standardized by the number of CFU and growth phase, as described previously [21]. Antimycobacterial activity was determined through the growth of the *M. tuberculosis* H37Rv culture on Middlebrook 7H9 liquid nutrient medium in the BACTEC MGIT960 automated system (BD, New Jersey, USA) in the presence of compounds at concentrations of 10, 20, and 40 µg/mL. *M. tuberculosis* H37Rv cultured on a medium containing no drugs (negative control), a medium with antituberculosis drugs at critical concentrations: rifampicin 1 μg/mL and isoniazid 0.1 μg/mL (positive control) and a medium with a solvent (solvent control). DMSO:H_2_O (30:70). This was added to the culture medium in a volume equal to the maximum volume of solvent added with the test concentrations of the compounds (monitoring the effect of solvent on culture growth). Each of the concentrations of the test compound, as well as control samples, was studied in triplicate. The level of bacteriostatic activity of the compound was evaluated by the method of proportions according to the principle described in the manual for BACTEC MGIT 960 [44]. If, on the day when the culture tubes were diluted, 1:99, 1:9, 1:3, and 1:1 showed growth of 400 GU and less than 100 GU were recorded in the test samples, then these concentrations of the compounds inhibited culture growth of at least 99%, 90%, 75% and 50%, respectively.

## Data Availability

The data presented in this study are available within the article.

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
