# Peer review of "New Flexible Analogues of 8-Aza-7-deazapurine Nucleosides as Potential Antibacterial Agents"

_ijms, 2023, doi:10.3390/ijms242015421_

Round 1

Reviewer 1 Report

In this manuscript the authors synthesized numerous compounds and evaluated them for antibacterial activity.  Four of the compounds demonstrated some antibacterial activity, although at quite high concentrations.

On lines 29, 166, and 203 “in 50 and 13 ug/ml” should read “at 50 and 13 ug/ml, respectively”.  Also, the authors state that these two compounds inhibited the growth of m. smegmatis, but do not indicate how much the growth was inhibited.  They should determine and report the concentration of compound that inhibits growth by 50% (IC50).

All concentrations are reported as “µg/ml”.  It would be better to report this data as “µM”.

On line140, “Scheme 1” should be “Scheme 2”.

Lines 174 and 175.  In the previous sentence the authors state that compound 20 ug/ml of compound 6 inhibited growth by 99%, and in this sentence the authors state that “However, compound 6 completely suppressed the growth of the culture” or 100% inhibition.  There is very little difference between 99% and 100% growth inhibition.  I doubt that the study was done with such precision to determine this type of difference.  Also, complete inhibition of growth does not give any information on the potency of the compounds.  It is possible that these compounds are very potent inhibitors of cell growth.  Again, the authors need to determine the IC50 of the compounds.

Inhibition of growth indicates that the compounds are cytostatic.  The authors should do experiments to determine whether or not the compounds are cytocidal. 

no comments

Author Response

The authors are grateful to the reviewer for the time spent and positive assessment of the manuscript. We appreciate all the reviewer’s comments. The answers to all questions and comments are below.

In this manuscript the authors synthesized numerous compounds and evaluated them for antibacterial activity.  Four of the compounds demonstrated some antibacterial activity, although at quite high concentrations.

On lines 29, 166, and 203 “in 50 and 13 ug/ml” should read “at 50 and 13 ug/ml, respectively”. 

Done

Also, the authors state that these two compounds inhibited the growth of m. smegmatis, but do not indicate how much the growth was inhibited.  They should determine and report the concentration of compound that inhibits growth by 50% (IC50).

The evaluation of MIC99 (or MIC100) rather than IC50 is customary in the field of antibacterial studies opposite to antiviral testing. For M.smegmatis mc2 155, as for other microbial samples growing for 1 or 2 days, at least 99% growth inhibition concentration in a liquid medium (MIC99) was determined. In this case, the MIC was determined by the absence of visible growth in the broth when the growth in the control was continued.

All concentrations are reported as “µg/ml”.  It would be better to report this data as “µM”.

It’s possible to report the measurements of antibacterial activity in µg/ml or µM. But we, as well as a lot of other investigators (see refs in the paper), traditionally use µg/ml. So we did the same this time to have an opportunity to compare data.

On line140, “Scheme 1” should be “Scheme 2”.

Done

Lines 174 and 175.  In the previous sentence the authors state that compound 20 ug/ml of compound 6 inhibited growth by 99%, and in this sentence the authors state that “However, compound 6 completely suppressed the growth of the culture” or 100% inhibition.  There is very little difference between 99% and 100% growth inhibition.  I doubt that the study was done with such precision to determine this type of difference.  Also, complete inhibition of growth does not give any information on the potency of the compounds.  It is possible that these compounds are very potent inhibitors of cell growth.  Again, the authors need to determine the IC50 of the compounds.

The main parameter MIC99, when testing the antibacterial activity of compound (6), was determined as 20 μg/ml. The same time, 100% growth inhibition was determined when the culture, exposed to the studied concentration of the compound 40 μg/ml, does not grow during the entire period of the experiment.

Inhibition of growth indicates that the compounds are cytostatic.  The authors should do experiments to determine whether or not the compounds are cytocidal.

The experiments by determination of the bactericidal concentration of compounds at this stage was not part of current study, because a large number of compounds were screened and so far we have done this for bacteriostatic activity. In the future, when the most active substances will be selected, we plan to study bactericidal activity.

The entire manuscript has been edited by our American colleague and co-author.

Reviewer 2 Report

Summary: The manuscript by Khandazhinskaya et al. describes the synthesis and evaluation of 8-aza-7-deazapurine nucleosides analogs. The author synthesizes these compounds via chemical synthesis and chemo-enzymatic methods and found that synthesis of analog 3 via both methods gave similar yields. The authors screened these analogs against M. smegmatis mc2 155 and M. tuberculosis H37Rv; where previously synthesized analog 19 and newly synthesized analog 9 was found active against M. smegmatis mc2 155 and compounds 6 and 10 were found active against M. tuberculosis H37Rv.

Review: This is a decent paper that offers insight into the synthesis via chemo-enzymatic pathway and chemical synthesis and test the analogs against antimicrobial activity. Overall, the work is an important contribution to the field of antibacterial drug development. The results are promising, and the novel flexible analogues show potential as antibacterial agents. However, there are few minor corrections that are listed below that should be addressed prior to publication.

1.     Line 50 can be rephrased for better clarity.

2.     In line 131 Yield mentioned is 14% whereas in line 346 yield mentioned is 18%.

3.     Figure 1, which shows previously synthesized analogs, has compound 13 redundantly assigned to two different molecules.

4.     Line 340 nomenclature of compound 6 should be bold.

5.     Line 102-103 and

Author Response

The authors are grateful to the reviewer for the time spent and positive assessment of the manuscript. We appreciate all the reviewer’s comments and added the answers.

Summary: The manuscript by Khandazhinskaya et al. describes the synthesis and evaluation of 8-aza-7-deazapurine nucleosides analogs. The author synthesizes these compounds via chemical synthesis and chemo-enzymatic methods and found that synthesis of analog 3 via both methods gave similar yields. The authors screened these analogs against M. smegmatis mc2 155 and M. tuberculosis H37Rv; where previously synthesized analog 19 and newly synthesized analog 9 was found active against M. smegmatis mc2 155 and compounds 6 and 10 were found active against M. tuberculosis H37Rv.

Review: This is a decent paper that offers insight into the synthesis via chemo-enzymatic pathway and chemical synthesis and test the analogs against antimicrobial activity. Overall, the work is an important contribution to the field of antibacterial drug development. The results are promising, and the novel flexible analogues show potential as antibacterial agents. However, there are few minor corrections that are listed below that should be addressed prior to publication.

  1. Line 50 can be rephrased for better clarity.

Done. The entire manuscript has been edited by our American colleague and co-author.

  1. In line 131 Yield mentioned is 14% whereas in line 346 yield mentioned is 18%.

The yield is 18%. Corrected.

  1. Figure 1, which shows previously synthesized analogs, has compound 13 redundantly assigned to two different molecules.

Compound 13 in first case put on Figure 1 by mistake, this is flex-base 1b.

  1. Line 340 nomenclature of compound 6 should be bold.

 Done

  1. Line 102-103 and

We apologize, somehow some of the text was removed. As mentioned above, the entire manuscript has been edited by our American colleague and co-author.
